# EVOLUTIONARY PERSPECTIVE ON MODEL FINE-TUNING

## ABSTRACT

Be it in natural language generation or in the image generation, massive performances gains have been achieved in the last years. While a substantial part of these advances can be attributed to improvement in machine learning architectures, an important role has also been played by the ever-increasing parameter number of machine learning models, which made from-scratch retraining of the models prohibitively expensive for a large number of users. In response to that, Transfer Learning (TL) - starting with an already good model and further training it on the data relevant to a new, related problem, gained in popularity. TL is formally similar to the natural evolution of genetic codes in response to shifting environment. Our core contribution, presented in this paper, is to define a class of evolutionary algorithms - Gillespie-Orr EA (GO-EA) and prove that they are equivalent in the limit to stochastic gradient descent (SGD). Based on this equivalence we present a number of tricks used y naturally evolving organisms to accelerate their adaptation, applicable to TL, as well as a set of hypotheses as to properties of artificial neural networks trained with SGD and GO-EA, resulting from such equivalence.

## 1 INTRODUCTION

Evolution-inspired algorithms are all but new in machine learning. Introduced in 1966, Simulated Evolution took the core components of the evolutionary processes as understood at the time - mutation and selection - and attempted simulate them in order to generate an artificial intelligence (Fogel et al., 1966). Appearing a mere 15 years after Robbins and Monro introduced the Stochastic Gradient Descent (SGD) (Robbins & Monro, 1951), Evolutionary Algorithms (EA) captured the imagination of the AI and ML communities and were refined to more closely follow the understanding of natural evolution, such as chromosomes and crossover (Schwefel, 1981), resulting in perhaps the most well-known evolutionary optimization and search algorithm of the family - the Genetic Algorithm (Goldberg, 1989).

However, this fascination slowly came to an end after in the late 70s and early 80, as several groups discovered independently that in most cases, SGD on the artificial neural networks (AANs) could achieve same, if not better, results as EA but with significantly lower computational expenses (Werbos, 1974; Parker, 1985; LeCun, 1985; Rumelhart et al., 1986). Shortly after, theoretical investigations in ANNs suggested that even simple architectures, such as multilayer feedforwards perceptrons, could work as universal approximators and be trained to approximate any measurable function, provided they had a sufficient number of layers, nodes per layer and non-linear activation functions (Hornik et al., 1989). A rapid flurry of innovations showing that ANNs armed with SGD could be scaled up to perform better and could efficiently deal with a number of problems considered hard until then (LeCun et al., 2015), switched the attention away from EA, until the superhuman performance of ImageNet on image classification tasks (Krizhevsky et al., 2012) made the ANN/SGD approach ubiquitous for computer vision, whereas the Transformer and its derivatives (Vaswani et al., 2017; Brown et al., 2020; Fedus et al., 2021) made ANN and SGD with inertia ubiquitous in Natural Language Processing.

However, despite its impressive success, the SGD has a flaw. By it's nature, it requires the manifold on which learning occurs to be differentiable. A limitation to which the Evolutionary Algorithms are not subject to, and one that is frequently encountered in conditions where multiple agents need to interact within a real or simulated environment (as for instance described by Majumdar et al.

(2020)). This advantage, combined with the advent of massively parallel computing that made the simultaneous evaluation of entire populations of models viable and led to a steady progress in EA application to ANNs, with the introduction and refinement of methods such as ESP (Gomez & Miikkulainen, 1997), CMA-ES (Hansen & Ostermeier, 2001), CoSynNE (Gomez et al., 2008), NES (Wierstra et al., 2008), reviewed in (Hansen et al., 2015). In addition to that, neuroevolution has been used to optimize the topology and hyperparametrs of ANNs before training them with SGD, as introduced by Floreano et al. (2008), or in combination with deep reinforcement learning (RL) (Mnih et al., 2015) - another approach to optimization in non-differentiable manifolds, as presented for instance by Fernando et al. (2017). Overall, the intersection of approaches trying to combine EA with ANN is a rapidly expanding area, reviewed more in depth by Galván & Mooney (2020).

However, compared to SGD, our understanding of how well EA are able to perform optimization and explore the parameter space of the machine learning model based on the data is limited. Specifically, it is unclear how prone they are to being trapped in local minima (Nguyen et al., 2021; Nguyen, 2019; Jacot et al., 2018), finding flat minima (Hochreiter & Schmidhuber, 1994; 1997; Goodfellow & Vinyals, 2015; Li et al., 2018), generalizing (Dinh et al., 2017; Zhang et al., 2020), memorizing (Arpit et al., 2017; Yun et al., 2019; Zhang et al., 2017; 2021) or interacting with ANNs architectures (Balduzzi et al., 2017; Li et al., 2018).

While the question in itself would likely be harder to approach than in the case of SGD, just because of the sheer diversity of EAs, we approach the topic specifically in the context of model fine-tuning, or transfer learning (Yosinski et al., 2014; Oquab et al., 2014; Bengio, 2012; Bengio et al., 2011; Caruana, 1995). The fine-tuning and transfer learning have grown in the importance over the last five years, given the exponential growth in the ANN models size, (Sanh et al., 2019; Brock et al., 2019; Fedus et al., 2021), going hand in hand with the training dataset sizes and computing power required to train them Brown et al. (2020). In this context, the from-scratch model training is prohibitively expensive for the vast majority of users, leading to the proliferation of model fine-tunes or transfers to closely related domains. As of the time of submission of this article, HuggingFace NLP model repository (HuggingFace, 2020) counts over 1300 model fine-tunes, of which around 700 of BERT model alone (Devlin et al., 2019).

Within the context of fine-tuning models or Tranfer Learning (Bozinovski & Fulgosi, 1976; Pratt, 1992; Pan & Yang, 2010), we show that thanks to very general results on the limit distributions derived by Gnedenko et al. (1968) and more generally Fréchet (1927); Fisher & Tippett (1928); Mises (1936); Gnedenko (1943), we can establish a direct equivalence between the SGD and the Fisher Geometric Model of Evolution (FGM) (Fisher, 1930) in the Orr-Gillespie formalization - a fairly large family of EA commonly used in theory of evolution (Tenaillon, 2014; Joyce et al., 2008; Orr, 2005).

By building upon this direct equivalence as well as some heuristics with regards to how much the FGM in the Orr-Gillespie formalization can be generalized outside its formal description (Joyce et al., 2008; Rokyta et al., 2008), we suggest additional insights on some features of ANNs trained with SGD, notably with regards to generalization and minima flatness, as well as implications for the EA family with which we drew the equivalence. We then offer a number of hypotheses applicable to ANNs trained with SGD, based on behaviors observed in biological system undergoing natural evolution.

## 1.1 FISHER GEOMETRIC MODEL IN THE ORR-GILLESPIE FORMALIZATION

The Fisher Geometric Model was introduced by sir Ronald Fisher as a cornerstone in his effort to unify the theory of evolution and reconcile Biometricists, convinced of the gradual evolution as presented by Darwin, and Geneticists, convinced of combinatorial genetic inheritance presented by Mendel (for a detailed review see Orr (2005)). The elegance of the model consisted in suggesting that the fitness, rather than being absolute, was a result of a mach between the organism and its environment, mediated by a set of traits that cold be adjusted to enable such an adaptation - aka a phenotypic space. Within this space, environment allowed the organism to achieve a maximal fitness in a single point, with the fitness gradually decaying as the organism moved further from the optimum. Gene alleles could either have a drastic effect on the phenotypic space, in which case their inheritance was Mendelian, or small, in which mixing of alleles from parents provided an illusion of gradual change (Fisher, 1930).

The resulting visual aid, presenting a stochastic walk towards the optimum on the fitness landscape, driven by the successive mutations and selections, is perhaps the most iconic aspect of the Fisher Geometric model and was declined for their own needs by fields with their own version of greedy exploration of fitness (or loss) landscapes.

However simplicity did not seem to bode well with the biological reality. The slow and gradual adaptation suggested by the model was at odds with fossil records (Gould & Lewontin, 1979) and couldn't be easily mapped to the recently discovered biological reality of the almost-binary-tape DNA sequence controlling the biological machinery of organisms and radically differences in phenotypes, back then almost always tracked to modifications in a single gene. Most importantly, the general and statistical nature of the FGM provided little insight into which genes were specifically under selection in organisms for different phenotypes

These shortcomings of the classical Fisher model, associated with a general change of perspective on the evolutionary genetics, led to a transition towards sequence substitution - oriented evolution. One of the first post-FGM models was the purely neutral drift model, where all the genetic and phenotype variation was a product of randomness and a number of successive bottleneck effects (Kimura & Crow, 1964; Kimura, 1968). With hard experimental evidence that some mutations were indeed advantageous and a large number was indeed deleterious, the model was refined to the near-neutral theory of evolution, where the vast majority of mutations had no effect on the fitness of the organism, a significant part had a deleterious effect, of which only a small fraction had a significant impact, and only a vanishingly small number of mutations were beneficial (Ohta, 1992).

However, the real breakthrough that allowed the FGM to be resurrected occurred through a paradigm shift in theoretical biology. First, a realization that fitness is environment-dependent. In other terms, organism that are highly adapted to their environment are subject to no evolutionary pressure and can remain unchangend for hundrends of millions of years, like the customary example of the horseshoe crab. An evolution will start only if a shift in fitness peak occurs , usually accompagniated by the initial population contraction allowing a rapid adaptive burst to occur (Lande, 1986), sometimes by leveraging pre-existing diversity in the genetic code space accumulated by neutral drift (Kauffman, 1969; Kauffman & Johnsen, 1991). Second, by leveraging recent results from limit distribution - specifically the Fisher-Tippet-Gnedenko (Fréchet, 1927; Fisher & Tippett, 1928; Mises, 1936; Gnedenko, 1943), closely realed to the generalization of the Central limit theorem by Gnedenko et al. (1968), Gillespie (1983; 1984) established that if the adaptative burst occurred under high selection/low mutation condition, starting from one already fit organism, the iterative steps in the genetic code space towards the most adapted genetic code would increase the fitness according to the Gumbel limit distribution. Consequently, Orr (2002; 2006) showed that the distribution of fitness of steps in the phenotypic space in the FGM model belonged to the same bassin of attraction as the distribution of fitness of steps in the genetic code space. In turn, it meant that the FGM could be used as a convenient visual model of evolutionary processes occurring in the genetic codes space, with concepts such as phenotypic dimension or fitness peak flatness becoming directly interpretable. Follow-up work by Joyce et al. (2008) showed that the conclusions reached for the distributions in the Gumbel limit distribution bassin of attraction still mostly held in the vicinity of it, in the bassins of attraction of Weibull and Fréchnet distributions.

Remarkably, the Gillespie-Fisher model of evolution is in no way linked to biological genetic codes specifically. It is applicable to any finite codes for which a fitness (or conversely loss) is defined and that are being iterative modified by a greedy search algorithm, starting from a code with already a high fitness value.

Fine-tuning real-world ANNs through SGDs is an instance of such code space search.

## 2 CENTRAL RESULTS

### 2.1 GILLESPIE-ORR EVOLUTIONARY ALGORITHM

In their formulation of the evolutionary process, Gillespie and Orr make three fundamental assumptions to make their model analytically tractable. Specifically:

- Haploid populations (single code evaluated for fitness)

- Under high selection ($Ns \gg 1$)

- In the low mutation limit ($N\mu < 1$)

Where $N$ is the population size, $s$ is a typical selection coefficient and $\mu$ is the per-site mutation rate.

The main purpose of those assumptions is to ensure that a new advantageous mutation swipes [1] through the population entirely upon appearance, before a next advantageous mutation can emerge and ensuring that a deleterious mutation never appears at the same time as an advantageous one.

While covering an important class of biological questions, such as drug resistance in cancers and bacteria Tenaillon (2014), this formulation is restrictive from the population genetics point of view. It is, however, perfectly adapted for the ANN neuroevolution, whenever after a mutation round only the highest fitness (conversely lowest loss) parameter $\theta$ is retained and no parameter mixing occurs. We will hence refer to such an Evolutionary Algorithm as *Gillespie-Orr Evolutionary Algorithm* (GO-EA)

## 2.2 SGD EQUIVALENCE TO THE GILLESPIE-ORR EA FOR A SUFFICIENTLY LOW LEARNING RATE

Let $f_{\theta}(\cdot)$ be a neural network parametrized by $\theta$, that maps inputs $\mathbf{X}$ of the form $\{\mathbf{x_i}\}_{\mathbf{i=1}}^{\mathbf{M}} \in \mathbb{Z}_{\mathbf{2}}^{\mathbf{n_x} \times \mathbf{d_x} \times \mathbf{M}}$ to outputs $\mathbf{Y}$ of the form $\{\mathbf{y_i}\}_{\mathbf{i=1}}^{\mathbf{M}} \in \mathbb{Z}_{\mathbf{2}}^{\mathbf{n_y} \times \mathbf{d_y} \times \mathbf{M}}$, where $\mathbb{Z}_2 = \{0, 1\}$, $d_x$ and $d_y$ are respectively the dimensionalitity of $x$ and $y$, $n_x$ and $n_y$ respectively the binary code length required to describe a single component of the vectors of $x$ and $y$ and $M$ the maximum number of inputs the network can encounter, with potentially $M = \inf$.

Let $\mathcal{L}_{\theta}$ be the fitness function associated to $f_{\theta}$ on the $\mathbf{X}$ and $\mathbf{Y}$. A priory, $\mathcal{L}$ is inaccessible, given it requires an evaluation on all the possible input-output pairs. However, it can estimated with a finite sample of inputs and outputs $\mathbf{X_{samp}}, \mathbf{Y_{samp}}$, giving us an $\hat{\mathcal{L}}_{\theta}|\mathbf{X}_{samp}, \mathbf{Y}_{samp}$.

Let $\mathcal{O}$ be a greedy optimization process, such that $\mathcal{O}(\theta) = \theta'$, with a rewrite capacity $d$, such that $||\theta' - \theta||_p < d$, where $p \in \mathbb{N}$ and $\hat{\mathcal{L}}_{\theta'}|\mathbf{X}_{samp}, \mathbf{Y}_{samp} \geq \hat{\mathcal{L}}_{\theta''}|\mathbf{X}_{samp}, \mathbf{Y}_{samp}$ for any $\theta''$ such that $||\theta'' - \theta||_p < d$.

Greedy optimization processes SGD and GO-EA are almost surely equivalent in the limit of SGD learning rate $l \to 0$ and GO-EA neighbourhood sample population $N \to \inf$ with GO-EA rewrite capacity $d = l$, up to a saddle point.

Since the SGD is applicable, we can perform the Taylor expansion in the neighbourhood of $\theta$ of $\hat{\mathcal{L}}_{\theta''}|\mathbf{X}_{samp}, \mathbf{Y}_{samp}$. As $d \to 0$, only the first order terms of the expansion remain, meaning that a gradient descent of the loss function $(-\hat{\mathcal{L}})$ with a step of $l$ will lead to the optimum within the $d$ ball around $\theta$. Given GO-EA has an infinite population, it will find the same optimum within the ball $d$. In case all the first order terms are nill, either the greedy optimization algorithm acheived a local minimum, in which case both SGD and GO-EA will stay put, or it is located in a local saddle point, in which case the SGD will acheive no movement and GO-EA will move to the highest fitness point within $d$ of $\theta$. In real conditions, the noise level provided by the sampling on the fitness function would lead the probability of the saddle point to vanish, if the samples on which $\hat{\mathcal{L}}$ is evaluated are different.

In order to achieve this result, we had to introduce the rewrite distance on the model parameters $\theta$, which operates on a norm and seemingly is incompatible with the single-mutation edits with which GO-EA operates in the context of genetic codes in biological organism. This is not the case. The parametrization $\theta$ combined with the ANN architecture is just one of the many possible ways to encode the model $f_{\theta}(\cdot)$, and it is certain that more compact and efficient codings exist, where the transition to a local minimum would correspond to single code character change.

---

[1]In population genetics and theory of evolution, an allele is said to swipe through the population when it prevalence increases until every single individual in the population has it. At this point, it is said to have been fixated in the population

### 2.3 Probability of finding better parameters during fine-tuning with Gillespie-Orr EA

In the context of fine tuning, we expect to start off a with a model $f_{\boldsymbol{\theta}_0}(\cdot)$ parametrized so that it already performs well on all the sample tests drawn from the distribution it was used to train with - aka $\forall(\mathbf{X}_{samp}, \mathbf{Y}_{samp}) \subset \mathbf{X} \times \mathbf{Y}, \mathbb{P}(\hat{\mathcal{L}}_{\boldsymbol{\theta}_0}|\mathbf{x}_{samp}, \mathbf{y}_{samp} \sim \max_{\boldsymbol{\theta}} \hat{\mathcal{L}}_{\boldsymbol{\theta}}|\mathbf{x}_{samp}, \mathbf{y}_{samp}) \sim 1$

Formally, fine-tuning consists in finding a new transfer parametrization $\boldsymbol{\theta}_T$, so that $\forall(\mathbf{X}_{samp}, \mathbf{Y}_{samp}) \subset \mathbf{X} \cup \mathbf{X}' \times \mathbf{Y} \cup \mathbf{Y}', \mathbb{P}(\hat{\mathcal{L}}_{\boldsymbol{\theta}_T}|\mathbf{x}_{samp}, \mathbf{y}_{samp} \sim \max_{\boldsymbol{\theta}} \hat{\mathcal{L}}_{\boldsymbol{\theta}}|\mathbf{x}_{samp}, \mathbf{y}_{samp}) \sim 1$, where the $\mathbf{X}'$ and $\mathbf{Y}'$ are new domains application of the model.

Assuming $|\boldsymbol{X}| >> |\boldsymbol{X}'|$ and $|\boldsymbol{Y}| >> |\boldsymbol{Y}'|$ (otherwise fine-tuning would be equivalent to model re-training), the model is already performing well on the fine-tuned model and the vast majority of the parameters within rewrite capacity $d$ of $\boldsymbol{\theta}_0$ would be deleterious or neutral, meaning that the parametrizations offering improvement would be distributed according to the generalized Pareto distribution (Pickands, 1975; Joyce et al., 2008), which in the case of Gumbel domain of attraction would result in an exponential distribution of fitnesses $\boldsymbol{s} = (s_1, ..., s_{i-1})$ where $s_j = \hat{\mathcal{L}}_{\boldsymbol{\theta}_j}|\mathbf{x}_{samp}, \mathbf{y}_{samp}$, the $j^{th}$ best parametrization of of better parametrizations and a probability to reach the better parametrization $\boldsymbol{\theta}_j$ of rank $j$ in the neighbourhood from a parametrization $\boldsymbol{\theta}_i$ of the rank $i$ of $\mathbb{P}_{i,j}(\boldsymbol{s}) = \frac{s_j}{\sum_{k=1}^{i-1} s_k}$

In other terms, with finite populations, GO-EA sampling the parametrization neighborhood of the current optimum $\boldsymbol{\theta}_i$ will find advantageous model code rewrites with the probability that's in reverse exponential probability of the difference between the loss associated to $\boldsymbol{\theta}_i$ and smallest possible loss within the edit distance budget.

While formally proven for the distributions in the Gumbel domain of attraction, this results has been shown to hold as well in the adjacent domains of attraction of the Weibull and Frechnet limit distributions, although the behavior further away from the Gumbel domain might differ radically, leading to all ranks for fitness being equally likely to be picked up in the limit case of Weibull bassin of attraction and only the lowest rank distribution being reachable by neighborhood sampling in the limit case of Frechnet bassin of attraction (reviewed in depth by Joyce et al. (2008)).

Unfortunately, the specific size of the sampling population N needed to sample at least one advantageous parameter within the rewrite capacity is directly connected to the effective latent dimension of the model - by analogy with the phenotypic space of the FGM. While we will discuss potential strategies to estimate it in the section 3, it is not directly accessible.

## 3 Implications of central results

### 3.1 Direct implications

#### 3.1.1 Robustness of the Gillespie-Orr Evolutionary Algorithm

A substantial amount of research has been invested to better understand how SGD interacts with ANNs architecture, managing to robustly find parametrizations for the ANNs that avoid local minima and provide reasonable noise-resistance and generalization capabilities.

To our knowledge, such results were entirely absent for the Evolutionary Algorithms until now. Thanks to the results in the 2.2, we can now claim that in case of a differentiable loss landscape, Gillespie-Orr Evolutionary Algorithm is equivalent to SGD in the limit of low learning rate and high sampling population size. This means that all the results previously shown for SGD are valid for GO-EA whenever this approximation holds.

#### 3.1.2 Computational advantage of SGD over EAs

Given these small learning rate and large sampling population approximation needed for the proof of 2.2, we can also fairly confidently say that whenever applicable, SGD is also more computationally efficient than GO-EA, given that the derivation and back-propagation are not computationally more expensive than new parameters sampling by more than the expected latent dimension of the model.

Given the simplicity of the GO-EA, we expect this computational efficiency relation to hold for other evolutionary algorithms, given that GO-EA is one of the possibly simplest ones.

### 3.1.3 MOST GENERALIZEABLE MODEL SELECTION AND EFFECTIVE EMBEDDING DIMENSION DETECTION

Prior work on diverse populations of biological systems evolving in a manner compatible with the Gillespie-Orr formalization has shown that it was possible to both estimate the effective phenotypic dimension, which for ANNs correspond to the latent dimension of the model, as well as to select within the population the sub-population that was the best at general performance (Kucharavy et al., 2018).

Specifically, in order to perform the latent dimension extraction, this work leveraged the Gnedenko-Kolmogorov formulation of the Central limit theorem, Gnedenko et al. (1968) that has proved that if a complex system can be altered in a large number of random ways, the resulting deviation from the base state converges to a Gaussian. Specifically, such a deviation would be visible along each axis relevant to adaptation to the environment, with the total deviation being characterized by the Chi-$n$ function, where $n$ is the effective phenotypic dimension of the organism/environment match, up to a renormalization. By subjecting this population to a number of diverse environments, it was possible to leverage the relationship between the mean and standard deviation of the fitnesses of the heterogeneous population among different population to both calculate the effective phenotypic dimension in which the population could move to match the environment, as well as the population that were the best at dealing with all the environments.

In the context of model fine-tuning, this could mean that in presence of a sufficiently diverse set of validation datasets and heterogeneous models, differing either by their architecture, initialization, hyperparameters or training dataset, it would be possible to both evaluate the effective latent dimension of the model family and the problem and to determine the model that is most likely to be a good starting point for fine-tunes that could cover all the datasets. Alternatively, the model ability to retain generality across validation datasets could also be used to identify models within the family least prone to catastrophic forgetting during the transfer learning.

## 3.2 HYPOTHESES

Whereas the previous subsection was dedicated to the direct implications of our central results, here we present several hypotheses based on the conceptual framework of evolutionary algorithms

### 3.2.1 MINIMA FLATNESS AS ERROR CORRECTION REDUNDANCY

SGD converging to flat minima is one of the conditions on the architecture of the ANN models for their training to be stable (Li et al., 2018).

Minima flatness was considered to the generalization abilities of the model through its presumed relationship to the minimal coding length of the model (Hochreiter & Schmidhuber, 1997; Goodfellow & Vinyals, 2015), although recently evidence to the contrary emerged (Dinh et al., 2017; Zhang et al., 2020; Mulayoff & Michaeli, 2020).

Within the theory of evolution, the flatness of the fitness peak is commonly associated to the tolerance to the neutral drift - aka error correction capabilities. By using this analogy, we suggest that just like in the context of the evolution, the flatness of the loss function minimum in ANNs optimized through SGD is determined by the redundnancy of the features used by the trained ANN to recognize patterns in the target data.

This intuition seems to be consistent with empirical observations about the loss function minima flatness. Architectures that provide the model with means to encode redundant features, such as with extremely large hidden layers or with skip-forwards connections in deep Convolutional Feed-Forwards ANNs, contribute to making the loss landscape minima more flat, as demonstrated by Li et al. (2018). Similarly, drop-out regularization Srivastava et al. (2014), forcing the ANNs to learn redundant, error-correcting codings seem to flatten minima as well, along with the smaller batches, which can contain a large proportion of samples that defy the heuristics that the ANN has learnt until now (Goodfellow & Vinyals, 2015)

From this perspective, we do not expect flatter minima to lead to better generalization, but rather to allow for more robust and less noise-sensitive models, which seems to be confirmed by the numerical experiments showing that ANNs with architectures, regularizations and training modes known to lead to flatter minima also tend to memorize less (Arpit et al., 2017).

### 3.2.2 FLAT MINIMA AND TRANSFER LEARNING

Building on top of the hypothesis presented above, if the minima flatness is indeed related to the classification robustness and error correction, we expect models that learnt a variety of error-correcting representations of training data to not be able to transfer those representations without training onto new data presenting similar features.

Intuitively, they rely on a simultaneous redundant subpaths through their ANN layers detecting redundant relevant features present in the training dataset. With only some of those features present in the dataset on which the transfer task is performed, their error correction property is likely to interfere with the the output of a corresponding output without an expected degree of redundant detection.

If this hypothesis is correct, a particular attention need to be payed when training ANNs that need to be both robustly map inputs to outputs in a noise-resistant manner and can be exposed to rare inputs presenting only some of the features on which the action need to be taken. We expect this problem to be separate from the adversarial examples one and more closely related to the generalization one, given that its goal is to recognize partial features.

## 4 HEURISTICS FROM THEORY OF EVOLUTION TO ACCELERATE THE FINE-TUNING PROCESS

Here we present a number of heuristics that are thought to be critical for the acceleration of the natural evolution, that we expect to be transferable to the training of well-formed ANNs through SGD.

### 4.1 MODEL MIXING IS NOT NECESSARY, ALTHOUGH CAN BE BENEFICIAL.

One of the prominent features of the later Genetic Programming compared to the early evolutionary algorithms was the emphasis put on mixing the models through "chromosome" "recombination". Directly mapping to the importance of the sexual reproduction in the context of natural evolution, it is neither necessary nor applicable in the context of GO-EA or SGD. Unlike in biological systems, there is no spurious mutation accumulation, so there is no need for the purifying selection to prevent Muller's ratchet from eliminating the population of ANNs (Lynch et al., 1993). Similarly, studies of model parameter interpolation between two good solutions indicate that the intermediate parameters tend to perform uniformly poorly without dedicated regularization (Goodfellow & Vinyals, 2015).

In case the model mixability is desireable however, for instance for rapid model aggregation attempts, or in the case the models independently drift away from the optimum, it is possible to develop regularization schemes that would preserve mixability (Livnat et al., 2008). Another reason such a mixability might be desirable, is that it might promote the diversity and redundance of feauture extraction subnets in the ANNs, further improving training stability and resilience to memorization.

### 4.2 RAPID EXPLORATION OF THE LATENT FEATURE SPACE

One of biological systems that are most consistent with the assumptions of the Gillespie-Orr model of natural evolution are pathogens developing resistance to treatments. While the means to achieve such resistance differ between the pathogens and treatments, they nonetheless use a common trick to accelerate their adaptation before going extinct. Specifically, rapidly generate random variation in the phenotypic space, in hopes that at least one of such variant populations would be more fit than the original one Kucharavy et al. (2018). This trick seems to be highly efficient, allowing the pathogens to adapt to new stressful conditions in a matter of a dozen of generations as opposed to thousands that would be expected in case of a gradual evolution. Given that the underlying model

for theoretical work of Kucharavy et al. (2018) is FGM with asexual reproduction, the results are fully compatible with GO-EA and hence with models fine-tuned with SGD.

To summarize the results from (Kucharavy et al., 2018), the dimensionality detection algorithms relies on a family of related models (for instance members of a population with random perturbation to parameters compared to a reference model), each evaluated on a heterogeneous benchmark. Based on the correlation of the average performance of models on each test task in the benchmark compared to the standard deviation of the models performance on each test, Kucharavy et al. (2018) shows that there is an expected correlation between the two and that a direct regression with two orthogonal parameters is possible to evaluate the underlying dimension of the problem. It is important to note that the dimension of the problem is not intrinsic to the problem at hand but also involves the architecture and parameter values of the ANN that has been trained to solve it, corresponding to the amount of independent "axes" along which the ANN can move to better adapt to task.

The mechanism we expect to limit the catastrophic forgetting requires a cross-evaluation of a family of models on a heterogeneous benchmark. Unlike the problem dimension evaluation, it evaluates the average performance of a model, as well as how uniformly it performs relative to other models on different tasks in the heterogeneous benchmark. Specifically, by calculating the Gini index of the model performance, Kucharavy et al. (2018) predicts that it will be inversely correlated with model performance and that the model with the lowest inequality of performance across tasks would also have reasonable performance. Given that it have a good average performance and perform well across most tasks in the benchmark, we expect that this model has not undergone catastrophic forgetting. It can be seen as a regularization that leverages intrinsic parallelizability of the GO-EA class of algorithms and ensures that as transfer learning is performed, at each exploration-selection cycle the loss of performance on other tasks in the benchmark is minimal while the new task is learnt. This regularization can also be applied in case of parallel model-fine tuning with SGD, resulting in a family of related models, from which the least "forgetfull" model is selected.

By analogy between the GO-EA and SGD adapted to fine-tuning the models, we expect that if a random variation was injected into the ANNs before starting the fine-tuning process, at the level where they would be getting close to leaving the flat minimum in their original loss space, their fine-tuning could be significantly accelerated, even if the least performing models are eliminated after the first couple of fine-tuning training epochs performed in parallel.

## 5 DISCUSSION

In this paper we establish a formal equivalence between the Gillespie-Orr model of evolution and SGD applied to ANNs. Build on top of strong limit distribution convergence results established by the Fisher-Tippet-Gnedenko theorem, GO-EA leverages the representation of ANNs as learnt codes that are greedily improved through neighbourhood exploration to provide an insight into how SGD might work and how it can be improved, as well as to what can be expected from parameter space search with GO-EA.

While we expect GO-EA algorithms to be more computationally expensive than SGD on differentiable manifolds, it does have two important application domains. First, in the cases where the evaluation of gradient and backpropagation of gradients are significantly more computationally expensive than the evaluation of the model performance (on the order of magnitude of the number of model parameters). In this circumstances, parallel computational capabilities and tricks allowing a more efficient communication of model updates, such as introduced in Such et al. (2017) can allow a faster and more computationally efficient model training. Second, in cases where the loss surface can be assumed to be smooth, but cannot be directly differentiated, such as in behavior strategies learning (as for instance described by Majumdar et al. (2020)).

While on the surface that last case seems to be rather limited, research in the context of SGD (Li et al., 2018) have shown that the smoothness of the loss landscape is dependent on the ANN architecture and parameter values rather than the problem alone. This suggests that there are classes of problems that area currently assumed to be difficult due to non-smooth loss landscapes that can be made more approachable by new ANN architectures and hence be efficiently explored by GO-EA algorithms.

Based on that formal equivalence we offer a new perspective on the minima flatness, which we link to the redundancy of the compressed codes representing the learnt model, rather than the minimal code length, as suggested previously, showing that is consistent with the experimentally observed results linking loss landscape flatness with skip connections, hidden layers width and the use of dropout regularization.

We further build on this insight in order to hypothesize that it is possible to identify the number of latent dimensions used by a model family to learn to map a training dataset inputs to corresponding outputs, as well as the most generalizeable model in a population, provided a sufficiently diverse set of secondary validation datasets is available.

Finally, we suggest a number of heuristics we expect would accelerate model fine-tuning or EA application to ANNs in general.

While the paper could greatly benefit from the experimental validation of hypotheses and heuristics presented here, the multiple model training restarts to collect statistics, hyperparameter space exploration and model population sizes needed for the EA methods mean that it is a task better suited for an entity with large computational capabilities and could represent a work in its own right.

In fact, ResNet-56 on CIFAR-10 used in (Li et al., 2018) to evaluate minima flatness requires 18 hours of training time on top-of-the line consumer hardware. An initial training run combined with fine-tuning is likely to multiply this time by a factor of magnitued, whereas orthogonal filter space search on a grid with filter normalization on the pre-fine-tuned model and fine-tuned models with intermediate steps would require similar a similar of magnitude of compute time. Combined with multiple restarts from different random seeds in order to obtain statistics on results, this means single experiment run times in the 70-80 days range of GPU time, assuming no other bottlenecks. While those experiments are trivially parallelizable, they require access to a cluster with sufficient compute budget to run them.

Moreover, our paper is consistent with prior experimental results, such as presented in Such et al. (2017). There, authors define a "genetic algorithm" that is fully elitist and does not proceed to any recombination and is hence an algorithm in the GO-EA class. By using this algorithm, they observe a number of features initially discovered and theoretically explored in the context of SGD, such as for instance local density of good solutions near a random initialized vector in case of sufficient model over-parametrization (Jacot et al., 2018). However, even a single training run of their model to train their 4 million parameter ANN for a single Atari game required 720 CPU core-hours for a single run - or 4-8 days of wall time on consumer-grade CPUs with 4-8 cores. Multiple restarts would be required to collect representative and comparable results, leading to simulation run times in the 20-40 days on consumer hardware.

We hope that our work provides novel perspective on the SGD convergence in the context of ANNs, as well as Evolutionary Algorithm application more generally.

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
