# OpenReview forum: "Evolutionary perspective on model fine-tuning"
_ICLR.cc/2022/Conference — ICLR 2022 Submitted_

### Official Review · Reviewer_5E1f · 2021-10-27

**Correctness:** 2
**Technical Novelty And Significance:** 3
**Empirical Novelty And Significance:** 1
**Recommendation:** 3
**Confidence:** 5

**Main Review:**

The paper is well argumented and potentially very interesting. I am myself a strong advocate of Black-Box Optimization and very excited for new work in the area. Writing aside however, the authors do not provide any semblance of experimentation or results sustaining their theoretical results, which is typically by itself unacceptable in major machine learning conferences. With still 1.5 pages from the page limits, I can see no problem for the authors to extend and complete this work.
The following points would need to be addressed in order for this work to be considered for publication:
- Either a theoretical or practical experiment, possibly both. For comparison of EC algorithms, particularly for continuous optimization, I would propose the standard BBOB/COCO. For the practical results, since neuroevolution is mentioned in depth, I would expect at least some standard RL control problem as found in the OpenAI Gym -- for simplicity, though any other environment or application from the literature would be acceptable, as long as the paper includes a comparison with previously published results.
- Many of the discussions proposed regarding EAs seem to utilize a traditional implementation, rather than the more modern implementations like CMA-ES (industry standard since over 15 years) or NES (itself over 10 years old), which utilize fitness ranking to sidestep entirely most of the problems addressed in this paper. I propose the experiments above should utilize standard CMA-ES as the baseline to showcase the advantages of this alternative approach.
- The key argument of the paper is the equivalence between SGD and Gillespie-Fischer EA / Gillespie-Orr EA. I think even a simple
- The paper does not provide a performance analysis of the newly proposed EAs. Given the comparison with SGD, and the unfortunately low performance of most ES implementations, I believe it is important to disclose whether the method could scale to the high dimensions and sample efficiency that we expect nowadays when discussing SGD.
- The otherwise extensive literature review first skips over the 20 years 1990-2010 in which Neuroevolution was the prolific state of the art for RL control, presenting it later in an incomplete manner. I would expect to see its discussion extended fairly in comparison to the other topics. Some of the main names include Hansen, Miikkulainen/Stanley, and for the current state of the art certainly Glasmachers.
- Neuroevolution is mostly applied to RL problem, given the compatibility of its fitness scoring system directly with a RL paradigm reinforcement (direct policy search). When equating the proposed ESs and SGD however, the authors hypothesize a differentiability of the model (though earlier stating that it is not a requirement in principle in BBO), and the availability of the error gradient itself. This introduces another big requirement, which is a supervised learning setup, with availability of labeled data. I find it necessary for improving exposition to split the cases explicitly, by (i) showing the equivalence between the proposed ES and SGD when subject to the same requirements of SGD, then (ii) motivating why would a user choose an ES when SGD is viable (as all studies so far point in the other direction), and most importantly (iii) what value is this equivalence in applications where SGD is not applicable (such as in problems with non-differentiable models or lack of labeled data), which is arguably the greatest strength of BBO methods such as ES.
- The literature review should also correctly attribute credit for Transfer Learning, called "fine-tuning" in the first part of the paper, which is a crucial component for some of the claims.
- The claim that the computational resources required for experimentation would be prohibitive should be supported by actual budgeting. As a counter-argument, I know I can run my own ES experiments on my own laptop; and a quick search online can find plenty of hobbyists solving complex problems without access to industry-grade computational resources. I suggest for example checking the [leaderboard for the OpenAI Gym](https://github.com/openai/gym/wiki/Leaderboard): problems such as the BipedalWalkerHardcore are extremely complex to solve, and yet there are evolved solutions proposed on the leaderboard that do not require high-performance computing. I point out as an example the results from `hardmaru` (David Ha), as they are commonly recognized as a high-quality approach. He used (2017) custom neural network code, a network with two layers of 40 neurons each, and the published python implementation implementation of CMA-ES: all the code is easily obtainable online and could run on a moderately recent laptop in a matter of days at worst.

I look forward to an extended version of this work as it certainly looks promising, and the research direction proposed is of high interest in the field.

**Summary Of The Paper:**

The paper presents the foundation for a new class of Evolution Strategies inspired from the work in biology of Gillspie and Orr published in the 1980s. Particularly intriguing are an equivalence with SGD (but limited to problems with the same assumptions and requirements of SGD) and heuristics to accelerate convergence using Transfer Learning. The paper however lacks any experimental validation of the theoretical results proposed.


**Summary Of The Review:**

Very interesting approach, incomplete comparison with the current state of the art (e.g. fitness ranking), and complete lack of any experimental results or any graphics. While I would not recommend the current state as ready for publication, I look forward to future more complete implementation of this work as it seems to be very promising.

---

> ### Author Response · Authors · 2021-11-23
> **Response to the reviewer 4**
>
> We thank the reviewer for their in-depth review and their interest in our approach.
>
> We note that along with other reviewers, the reviewer sees the experiments as important to the demonstration of the interest of the paper. We provide the evaluation of the computational budget required for the experiments evaluating our hypotheses with respect to the minima flatness in our response to the reviewer 1. We also note that the examples of validation by David Ha on the BipedalWalkerHardcore task, according to the paper describing it (assumed [1]), required "40h on 96 CPU machine" (footnote 2), due to the multiple restarts with different random seeds. On a modern CPU with 4-8 cores, this translates to 20-40 days of runtime. As we mention in our responses to the reviewer 1, flatness hypothesis validation adds additional overhead to that. In case we wrongly identified the algorithm and the evaluation process, we would be thankful for specific pointers.
>
> We thank the reviewer for providing additional references for the advances in the field of the neuroevolution as well as lack or proper attribution of the Transfer Learning and modify the paper to address these issues.
>
> We are unsure to understand the reviewer’s claim that CMA-ES or NES algorithms sidestep problems faced by the GO-EA class. In our understanding, CMA-ES and NES rely on the natural gradient estimation. To our understanding, this does not resolve the fundamental problem of curse of dimensionality in modern highly-overparametrized ANNs and introduces additional challenges. Notably, the modification of the search space according to the natural gradient spreads the search volume, meaning a less dense sampling and potentially missed saddle point bifurcations towards better optima in non-convex problems - a problem considered as more serious than local minima traps in SGD. Besides, a minimum along an axis is not expected to give an access to the optimum in non-convex problems, rather changing the context and leading to a movement in a different direction to become adaptive. CMA-ES and NES seem to require a convergence along one axis before focusing on a different axis, rather than continuously following the local gradient like SGD and GO-EA do. Finally, from the computational perspective the natural gradient estimation adds an additional communication and computation overhead due to the natural gradient, denying the trivial parallelizability advantage of EAs over RL, as well as making impossible network compression tricks that allowed scaling of GO-EA algorithms to ANNs with millions of parameters, such as random seed communication proposed by [2] – which to our knowledge are the only successful EAs on large models that outperformed RL approaches.
>
> Moreover, we once again insist that the core result of our paper is the proof of equivalence of GO-EA and SGD in the limit rather than a proposal of a new algorithm. Specifically, our aim is to enable the application of insights and mathematical tools developed in the context of SGD to the problems where a locally smooth loss surface is expected but is not directly accessible for differentiation. Given the counter-intuitive behavior of ANNs with a large number of parameters, such as minima connexity [3] or local density of good solution in overparametrized moels [4] (observed by [2]), we expect such insight transfers to be useful to understand and improve current EAs.
>
> While the locally smooth surfaces seem to be a limited class of the problem, work on the loss surface smoothness in the context of SGD shows that it depends not only on the nature of the problem, but also a structure of the ANNs itself [5], suggesting that hard problems can be made accessible by changing ANN architectures and regularizations, at least when trained with GO-EA class of algorithms (a result also empirically observed by [2]). As we mention in the reviewed paper, unless the gradient computation is harder than evaluation of a single solution by the order of magnitude comparable with the number of parameters of model, we expect the SGD to perform better whenever it is applicable
>
> We further clarify this point in the revised draft and thank the reviewer for bringing them to our attention.
>
>
> Referneces:
>  [1] Artificial Life (2019) Reinforcement Learning for Improving Agent Design - https://arxiv.org/abs/1810.03779
>
>  [2] CoRR (2017) Deep Neuroevolution: Genetic Algorithms Are a Competitive Alternative for Training Deep Neural Networks for Reinforcement Learning - https://arxiv.org/pdf/1712.06567.pdf
>
>  [3] CoRR (2021) When Are Solutions Connected in Deep Networks? - https://arxiv.org/abs/2102.09671
>
>  [4] NeurIPS (2018) Neural Tangent Kernel: Convergence and Generalization in Neural Networks - https://proceedings.neurips.cc/paper/2018/file/5a4be1fa34e62bb8a6ec6b91d2462f5a-Paper.pdf
>
>  [5] NeurIPS (2018) Visualizing the Loss Landscape of Neural Nets - https://proceedings.neurips.cc/paper/2018/file/a41b3bb3e6b050b6c9067c67f663b915-Paper.pdf

---

> > ### Comment · Reviewer_5E1f · 2021-11-30
> > **Answer to comment**
> >
> > Thank you for your response and edits. I understand that this paper is theoretical, and there are conferences where purely theoretical papers are well received. There are also conferences where a purely experimental approach is sufficient for publication. I believe acceptance into a top venue such as ICLR requires the highest effort, with both theoretical and practical sides being validated.
> >
> > Your comment explains why this is hard, not why this has not (or should not have) been done. I myself have run experiments for 15 days on a 4-core CPU, albeit that was admittedly that was almost 20 years ago. If this is still today all the computational availability at your hosting institution, it still should allow you to run Separable CMA-ES, SNES or even LM-MA-ES in a matter of hours, with results that I still expect would support my original thesis. Even more, with preliminary results I would expect it to be easy to find a student at another institution who is willing to join a collaboration and run the experiments for your work, fostering networking and collaboration, and producing a much higher quality publication.
> >
> > I would like to propose as an example of the reasons why we hold to such high standards: your comment on CMA-ES and NES. Both you and I are experts in the topic with over a decade of experience (I am assuming, given your emphasis), yet we disagree on that point. If you are indeed correct (and I wish you are for the sake of the field over my own opinion), there is one simple way to show your correctness: run the experiments, use them as a baseline, and show the improvements from your method. I look forward to the results of this evaluation and as I reiterate my belief that these results can be significantly impactful in the field if validated, I wish you in all sincerity good luck.

---

### Official Review · Reviewer_DYJE · 2021-10-29

**Correctness:** 2
**Technical Novelty And Significance:** 2
**Empirical Novelty And Significance:** Not applicable
**Recommendation:** 3
**Confidence:** 4

**Main Review:**

Strengths

The authors provided a significant amount of generalization in their methodology using the Gillespie-Orr genetic algorithm to be useful outside of the specific application they described.
In proving the equivalence between the genetic algorithm and SGD, they provided specific details on the proof and limitations that existed therein. It provides a pathway for the expansion of performance approximation of GO-EA given the described equivalence holds.
Within the description of the minima flatness as related to a genetic algorithm’s error correction, they provided a well detailed analysis on how robust, noise-resistant models might fail to transfer successfully as the redundant features will interfere with the capability of the model to train under the new dataset. In their apt analogy, this would be the population of an EA suppressing beneficial changes due to strong error correction in the original population.
Finally, the two heuristics proposed for the improvement of the model fine-tuning process provided a great description of the problem space each feature of the GA would be able to tackle, and how it was related.
Overall, the detailed descriptions of the evolutionary processes and their relations to model fine-tuning processes provided a very convincing argument for the usefulness of their ideas.

Weaknesses

This paper mentions multiple shortcomings on the usefulness of genetic algorithms in the field of fine-tuning and on their specific equivalency between the Gillespie-Orr GA and SGD. Overall, the authors provided some details but failed to provide an in-depth analysis on the equivalency given their assumptions failed.
The section on dimensionality detection was too brief to discuss how they calculated the effective phenotypic dimension and leveraged it to find the best population. While including this information is still useful, the paper in general would benefit greatly to discuss how some of the discussed facets of using a genetic algorithm would look like. For instance, discussing the specific relationship between the newly found best population would be able to reduce catastrophic forgetting would help improve the credibility of this statement.
In general, as mentioned at the end of the paper as a flaw, this paper would benefit greatly from empirical evidence supporting the claims made. Without it, many of the claims lack credibility as the proposed gain in accuracy or generality in models might amount to very little in practice.

This paper contains a number of minor spelling and syntax errors that would benefit from an extra review.
Also, although grabbing empirical data is likely outside the scope of this project, I feel that the addition of more information into some of the secondary points brought up in the paper would provide a more comprehensive view on the outcome of the original goal of the paper.

Overall, the lack of empirical results and more in-depth theoretical analysis severely limits the significance of the paper.

**Summary Of The Paper:**

The authors of the paper sought to describe the relationship between the Gillespie-Orr model of evolution and Stochastic Gradient Descent in an attempt to describe the usefulness of evolutionary processes from a model fine-tuning perspective. After first describing the general use and limitations of the two methods, they proved an equivalency between the two and the improved ability of Gillespie-Orr EA to find better model parameters during fine tuning. In addition, they constructed a set of two potential cases for the use of evolutionary algorithms in the general methodology of model fine-tuning as a potential performance improvement.
This paper addresses a novel concept within the field of model fine-tuning and provides information on its practicality and usefulness.

**Summary Of The Review:**

A nice investigation with limited significance.

---

> ### Author Response · Authors · 2021-11-23
> **Response to the reviewer 3**
>
> We thank the reviewer for their review and their interest in our approach.
>
> We thank the reviewer for attracting our attention to the spelling and syntax errors and are currently proof-reading the paper to remove them and improve overall readability.
>
> We note the reviewer's suggesting that experiments supporting our hypotheses would make our paper stronger, in agreement with other reviewers, and attract their attention to the explanation provided to the reviewer 1. Once again, we would welcome any suggestions for experiments reasonable computational budget that the reviewer would consider as a satisfactory evaluation of our hypotheses; as well as draw the attention of the reviewer to the fact that the core result of the paper is theoretical and stands as is, already allowing valuable insight.
>
> We would like to draw the attention of the reviewer as well to the fact that the limit conditions used for proof of equivalence between GO-EA and SGD are not expected to occur in cases of deployment of algorithms in the GO-EA class. Instead, the limit conditions are a theoretical tool that allows to apply the insights and mathematical tools developed in the context of SGD to EAs in the GO-EA class and vice-versa. We provide an additional example of such application in in response to reviewers 1 and 2, where authors of an experimental work [1] define an evolutionary search algorithm in the GO-EA class and stumbled across empirical results consistent with theory developed in the context of SGD [2].
>
> We note the reviewers' request for a more detailed description of dimensionality detection algorithm and of the mechanism allowing to limit catastrophic forgetting. These algorithms have been developed in the context of biological evolution within the framework of Fisher's Geometrical Model [3]. Our contribution consisted in noticing that the assumptions used to develop those algorithms are satisfied for the GO-EA class algorithms and hence applicable to the families of ANNs trained with GO-EA or SGD. Because of that, we minimized our discussion with regards to this section to avoid reproducing as-is equations from [3].
>
> Specifically, the dimensionality detection algorithms rely on a family of related models (for instance members of a population with random perturbation to parameters compared to a reference model), each evaluated on a heterogeneous benchmark. Based on the correlation of the average performance of models on each test task in the benchmark compared to the standard deviation of the models’ performance on each test [3] shows that there is an expected correlation between the two and that a direct regression with two orthogonal parameters is possible to evaluate the underlying dimension of the problem. It is important to note that the dimension of the problem is not intrinsic to the problem at hand but also involves the architecture and parameter values of the ANN that has been trained to solve it, corresponding to the number of independent "axes" along which the ANN can move to better adapt to task.
>
> The mechanism we expect to limit the catastrophic forgetting requires a cross-evaluation of a family of models on a heterogeneous benchmark. Unlike the problem dimension evaluation, it evaluates the average performance of a model, as well as how uniformly it performs relative to other models on different tasks in the heterogeneous benchmark. Specifically, by calculating the Gini index of the model performance, [3] predicts that it will be inversely correlated with model performance and that the model with the lowest inequality of performance across tasks would also have reasonable performance. Given that it has a good average performance and perform well across most tasks in the benchmark, we expect that this model has not undergone catastrophic forgetting. It can be seen as a regularization that leverages intrinsic parallelizability of the GO-EA class of algorithms and ensures that as transfer learning is performed, at each exploration-selection cycle the loss of performance on other tasks in the benchmark is minimal while the new task is learnt. This regularization can also be applied in case of parallel model-fine tuning with SGD, resulting in a family of related models.
>
> We modify the draft to clarify these points and thank the reviewer for bringing them to our attention.
>
> References:
>
>  [1] CoRR (2017) Deep Neuroevolution: Genetic Algorithms Are a Competitive Alternative for Training Deep Neural Networks for Reinforcement Learning - https://arxiv.org/pdf/1712.06567.pdf
>
>  [2] NeurIPS (2018) Neural Tangent Kernel: Convergence and Generalization in Neural Networks - https://proceedings.neurips.cc/paper/2018/file/5a4be1fa34e62bb8a6ec6b91d2462f5a-Paper.pdf
>
>  [3] MBoC (2018) Robustness and evolvability of heterogeneous cell populations - https://doi.org/10.1091/mbc.E18-01-0070

---

> > ### Comment · Reviewer_DYJE · 2021-11-29
> > **Thank you**
> >
> > Thank you for your reply! Regarding the experiments, I believe that anything in line with the current literature on hyperparameter optimization would be acceptable, or even much smaller-scale experiments if they are convincing.

---

### Official Review · Reviewer_UoCs · 2021-11-02

**Correctness:** 3
**Technical Novelty And Significance:** 2
**Empirical Novelty And Significance:** 2
**Recommendation:** 3
**Confidence:** 4

**Main Review:**

Strengths
- This paper reviews a long history of evolutionary and stochastic gradient descent algorithms.
- It provides a new perspective that the fine-tuning process is similar to the adaptation in environmental change.

Weaknesses
- Their central algorithm is from previous milestone works.
- The perspective of evolving the parameter itself is not new.
  - Recent works apply evolutionary algorithms directly into the parameters of ANN. Here I reference some works in the reinforcement learning field as an example [1-5].
- The central results, hypothesis, and heuristics are not supported by the experimental results.
  - It is better to show some results compared with SGD.
  - Further analysis is necessary to provide the validity of claimed heuristics.
- Several typos and not organized sentences.
  - Some repeated words, like "model model", "steps steps" and more.
  - The sentence is not finished, "... in reverse exponential probability of the"
  - Some paragraphs are extremely long that makes it hard to read, and some are very short, seems no reason to separate

References
1.  Salimans, Tim, et al. "Evolution strategies as a scalable alternative to reinforcement learning." arXiv preprint arXiv:1703.03864 (2017).
2.  Such, Felipe Petroski, et al. "Deep neuroevolution: Genetic algorithms are a competitive alternative for training deep neural networks for reinforcement learning." arXiv preprint arXiv:1712.06567 (2017).
3.  Khadka, Shauharda, and Kagan Tumer. "Evolution-guided policy gradient in reinforcement learning." Proceedings of the 32nd International Conference on Neural Information Processing Systems. 2018.
4. Pourchot, Aloïs, and Olivier Sigaud. "CEM-RL: Combining evolutionary and gradient-based methods for policy search." arXiv preprint arXiv:1810.01222 (2018).
5. Lee, Kyunghyun, et al. "An Efficient Asynchronous Method for Integrating Evolutionary and Gradient-based Policy Search." arXiv preprint arXiv:2012.05417 (2020).

**Summary Of The Paper:**

This work introduces Gillespie-Orr Evolutionary Algorithm (GO-EA) and applies it to the fine-tuning of parameters. It shows the perspective of evolution on model fine-tuning by comparing with SGD and analyzing the probability of finding better parameters. Finally, it provides some heuristic techniques to accelerate the fine-tuning process.


**Summary Of The Review:**

Although it provides a new perspective in fine-tuning process, the core idea seems not novel and is not supported by robust experiments.
Also, the writing seems not thoroughly reviewed.

---

> ### Author Response · Authors · 2021-11-23
> **Response to reviewer 2**
>
> We thank the reviewer for their review and their interest in our approach.
>
> We would like to clarify that while our work indeed builds on prior work in theory of evolution, notably on Allen Orr's and John Gillespie's explanation as to why insight from simple geometrical toy model (Fisher's geometrical model of evolution) had any predictive power for the evolution of the genetic sequence of organisms undergoing adaptation to complex environments.
>
> However, the definition of the GO-EA class of algorithms and the proof of their equivalence to SGD in a limit are novel and constitute the core of our contribution presented in this paper. To the best of our knowledge, while algorithms belonging to the GO-EA class have been described and used previously, the class in itself is novel, with the closest related evolutionary class being the (1+λ)-ES class of evolutionary search algorithms, when the neutral drift in GO-EA model is assumed to be null [1]. However, the neutral drift is key to the stochasticity of GO-EA class and allows the equivalence to the *stochastic* gradient descent. Moreover, neutral drift is currently understood as being key to sustained evolution [2] and has been proposed, in the quasi-species theory, to be the process by which living organisms search for genetic encoding of phenotypic features that is resilient to small perturbation - aka "fitness peak flatness" [3]. We attempt to clarify this point in the revised draft.
>
> We thank the reviewer for providing interesting references for the application of evolutionary algorithms directly to the ANNs. We believe we mention the application of evolutionary algorithms to ANNs in the introduction by citing a review on the subject, but to avoid the confusion, we will add the papers suggested by the reviewers.
>
> In fact, one of the papers suggested by the reviewer - specifically [4] - provides us with an excellent illustration of the interest of our work. As we mention in our response to the reviewer 1, despite naming their algorithm "Genetic Algorithm", the absolute elitism and absence of recombination make the evolutionary algorithm defined in [4] a member of the GO-EA class. Quite interestingly, while trying to understand the performance of the evolutionary algorithm compared to other methods, they stumble across the "good solutions can be found within a short edit distance from random initial parameters" property, that has been recently a focus of the theoretical studies in the SGD context (cf notably [5]). Our work suggests that rather than a fluke, this result is expected in case of a sufficient model overparameterization and locally smooth loss landscape and can be approached with tools developed in the context of SGD, given that the passage to the limit we use in our proof holds in that context.
>
> We note the reviewer's suggestion that our work could greatly benefit from computational experiments supporting our hypotheses, which aligns with the suggestion from other reviewers. Unfortunately, the validation of the hypotheses relative to the minima flatness and error redundance in the same manner as was achieved by the landmark publications in that domain is outside the capabilities, we currently have access to. We would be thankful for any suggestions as to possible experimental approaches that would allow us to bypass the computational cost of the evaluation, but we would also like to attract the attention of the reviewer to the fact that the core result of the paper is theoretical and stands as is, already allowing valuable insight.
>
> We thank the reviewer for highlighting the issues with repeated words, unfinished sentences and excessively long paragraphs - we are proof-reading and editing the paper to remove such errors and improve readability.
>
>
> References:
>
>  [1] Springer Handbook of Computational Intelligence (2015) Evolution Strategies -  https://doi.org/10.1007/978-3-662-43505-2_44
>
>  [2] Journal of Theoretical Biology (1987) Towards a general theory of adaptive walks on rugged landscapes - https://doi.org/10.1016/s0022-5193(87)80029-2
>
>  [3] PNAS (1999) Neutral evolution of mutational robustness - https://doi.org/10.1073/pnas.96.17.9716
>
>  [4] CoRR (2017) Deep Neuroevolution: Genetic Algorithms Are a Competitive Alternative for Training Deep Neural Networks for Reinforcement Learning - https://arxiv.org/pdf/1712.06567.pdf
>
>  [5] NeurIPS (2018) Neural Tangent Kernel: Convergence and Generalization in Neural Networks - https://proceedings.neurips.cc/paper/2018/file/5a4be1fa34e62bb8a6ec6b91d2462f5a-Paper.pdf

---

> > ### Comment · Reviewer_UoCs · 2021-11-30
> > **Answer to comment**
> >
> > I understand that this paper is theoretical. However, I still think that at least a simple experimental result should be included for validation.

---

### Official Review · Reviewer_DWHG · 2021-11-02

**Correctness:** 2
**Technical Novelty And Significance:** 3
**Empirical Novelty And Significance:** 3
**Recommendation:** 5
**Confidence:** 4

**Main Review:**

The comparison of the SGD and the evolutionary algorithm is quite interesting but not completely new from the conceptual point of view. The paper however introduces a very specific - weaker type - of evolutionary algorithm that is easier to be formalized. The main conclusions of the paper are that the flatness of the SGD minima is an equivalent statement for the stability learning in ANN. Another hypothesis proposed is that the transfer learning of a model with SGD minima flatness can only be transferred to a data that would result in a similar minima flatness.

While the hypothesis seems to be plausible there is a general lack of formalism which prevents to clearly discussing the validity of the model.  The generality of the proposed hypotheses would require at least experimental verification in order to provide supporting evidence from the data stand point of view.

For instance, the notion of the SGD minima flatness is not well enough defined to make decision what kind data is required so that the transfer learning is successful. Similarly the hypothesis of the error correcting ability of ANN, while seems empirically correct needs to be verified.

**Summary Of The Paper:**

This paper introduces a conceptual equivalence between a certain class of evolutionary algorithms and the stochastic gradient descent. The paper is relatively well written but often introduces notation without proper definitions.

**Summary Of The Review:**

The paper in general lacks supporting formulation that would allow clear understanding and clear conclusions about the validity of the proposed statements.

---

> ### Author Response · Authors · 2021-11-23
> **Response to the reviewer 1**
>
> We thank the reviewer for their review and their interest in our approach.
>
> While we indeed introduce a more restricted class of evolutionary algorithm, this class is able to out-perform current reinforcement learning approaches and, to our knowledge, is the only one to have been applied to large ANNs [1] (4 million parameters in that paper). The algorithm in [1] is a member of a GO-EA, given the absolute elitism and lack of recombination. Quite interestingly, authors of [1] empirically discovered the "dense search in the local neighborhood provides good solutions" property that had been discovered and extensively studied in the context of SGDs [2].
>
> Our proof of equivalence between algorithms in the GO-EA class and the SGD explains that the observation made by authors of [1] is not spurious, but instead can be understood using tools developed in [2].
>
> It is an example of application of the proof of equivalence - which is the central result of our paper itself and is a new result, to the best of our knowledge. The hypotheses provided, such as flatness of minima relation to the redundancy of codes learnt during the training by SGD, or with regards to the minima flatness evolution during transfer learning are derived from that result. With regards to the minima flatness definition, we use the standard definition cited in [3] - stability of the loss upon a random parameter perturbation, with per-filter normalization to account for the scale-invariance of common neural network architectures. We are not aware of any other definitions in use and would be thankful for any alternative definitions that would allow us to refine our formulation.
>
> As we mention in the conclusion of our paper, an experimental validation of the hypothesis would require multiple pre-training and fine-tuning runs of the model, with pauses at intermediate checkpoint in order to experimentally evaluate the flatness of minima at the starting point, end point and on the path linking the two, in order to get sufficient statistics as to difference in flatness. While trivially parallelizable, this approach requires either access to a compute cluster or extensive time. While a single run of ResNet-50 pre-training on a top-of-the-line consumer graphic card would need over 18h, with similar time for a single model fine-tune, the evaluation of the flatness of the model requires a computation of filter-normalized random perturbations in a grid at the starting, final and intermediate fine-tuning states for each model run. By comparison, [3] used ResNet-56 variants trained on CIFAR-10 in their experiments and computed the flatness only at the end of the run.
>
> Moreover, the study of the GO-EA equivalence would require parallel evaluation of large populations at every mutation-selection step. [1] suggests that for 4 million parameters, samples of ~ 1000 individuals were effective. While the task is embarrassingly parallel and allows for rapid exploration in presence of large parallel computation capabilities, the total wall time of even a single run is significantly higher than the one of SGD.
>
> It is important to mention that this does not limit the applicability of the GO-EA. SGD is more efficient on differentiable surfaces, however in cases where the loss surface can be assumed to be locally smooth but not directly differentiable, or when the loss evaluation is significantly faster than differentiation and back-propagation and significant parallel compute capabilities are available, GO-EA class algorithms become a compelling alternative. While the case of locally smooth but non-differentiable loss functions appears to be limiting, results in SGD ([3]) seem to indicate that the loss function smoothness is a property of ANN architecture as well, rather than of the problem alone, and is likely to be achievable by modifying the ANN architecture in a number of problems currently assumed to be non-smooth.
>
> In response to this review and others, we are adding the estimation of computational expenses into the draft. We would be thankful for any recommendations of simpler tasks that would allow us to illustrate our hypotheses within available computational capabilities.
>
> However, we would also like to attract the attention of the reviewer to the fact that the core results of the paper are theoretical and stand by themselves as is, already allowing insight into the behavior of ANNs trained using GO-EA class of algorithms.
>
> References:
>
>  [1] CoRR (2017) Deep Neuroevolution: Genetic Algorithms Are a Competitive Alternative for Training Deep Neural Networks for Reinforcement Learning - https://arxiv.org/pdf/1712.06567.pdf
>
>  [2] NeurIPS (2018) Neural Tangent Kernel: Convergence and Generalization in Neural Networks - https://proceedings.neurips.cc/paper/2018/file/5a4be1fa34e62bb8a6ec6b91d2462f5a-Paper.pdf
>
>  [3] NeurIPS (2018) Visualizing the Loss Landscape of Neural Nets - https://proceedings.neurips.cc/paper/2018/file/a41b3bb3e6b050b6c9067c67f663b915-Paper.pdf

---

> > ### Comment · Reviewer_DWHG · 2021-11-29
> > **Answer to Comment**
> >
> > Yes I understand that it is theoretical paper. But it is formulated more like a set of observations rather than a generalization theory. Now the observations seems to be very valid but to finish the paper either some concept experiment or a formalization would do the thing and close it properly

---

### Author Response · Authors · 2021-12-09
**Overall response to the reviewers**

We thank the reviewers for their feedback, that contributed to improving the paper.

At this stage, the reviewers seem to have reached consensus that while theoretical approach is novel and significant to the field, experimental validation is still essential to support the main claims and provide empirical support to presented hypotheses.

While we believe our contribution stands as is, we understand the reviewers' demand for experimental validation and are currently investigating how it can be feasibly accomplished.

---

### Decision · Program_Chairs · 2022-01-20

**Decision:**

Reject

**Comment:**

The reviewers agree that this is an interesting treatise on some relationships between SGD fine tuning and evolutionary algorithms. All reviewers have requested some experimental validation or demonstration of the theory developed in this paper, which is not currently included. Whilst the computational requirements (and time required) may be long, this will significantly assist the many readers of the paper and save them from having to run such an experiment many times themselves.  The reviewers provided a number of suggestions of how this might be done. The reviewers also highlighted a number of specific improvements that can be made to the writing of the paper.